# Engineered Phage Endolysin Eliminates *Gardnerella* Biofilm without Damaging Beneficial Bacteria in Bacterial Vaginosis Ex Vivo

**DOI:** 10.3390/pathogens10010054

**Published:** 2021-01-08

**Authors:** Christine Landlinger, Lenka Tisakova, Vera Oberbauer, Timo Schwebs, Abbas Muhammad, Agnieszka Latka, Leen Van Simaey, Mario Vaneechoutte, Alexander Guschin, Gregory Resch, Sonja Swidsinski, Alexander Swidsinski, Lorenzo Corsini

**Affiliations:** 1PhagoMed Biopharma GmbH, Vienna Biocenter, 1110 Wien, Austria; christine.landlinger@phagomed.com (C.L.); lenka.tisakova@phagomed.com (L.T.); vera.oberbauer@phagomed.com (V.O.); timo.schwebs@phagomed.com (T.S.); abbas.muhammad@phagomed.com (A.M.); 2Laboratory Bacteriology Research, Department of Diagnostic Sciences, Faculty of Medicine & Health Sciences, Ghent University, Flanders, 9000 Gent, Belgium; Agnieszka.Latka@UGent.be (A.L.); Leen.VanSimaey@UGent.be (L.V.S.); Mario.vaneechoutte@ugent.be (M.V.); 3Department of Pathogen Biology and Immunology, Institute of Genetics and Microbiology, University of Wroclaw, 51-148 Wroclaw, Poland; 4Moscow Scientific and Practical Center of Dermatovenerology and Cosmetology Moscow, 119071 Moscow, Russia; alegus65@mail.ru; 5Department of Fundamental Microbiology, University of Lausanne, 1015 Lausanne, Switzerland; gregory.resch@unil.ch; 6MDI Limbach Berlin GmbH, 13407 Berlin, Germany; Sonja.Swidsinski@mvz-labor-berlin.de; 7Medizinische Klinik, Charité CCM, Humboldt Universität, 10117 Berlin, Germany; alexander.swidsinski@charite.de; 8Institute of Molecular Medicine, Sechenov First Moscow State Medical University, 119435 Moscow, Russia

**Keywords:** bacterial vaginosis, *Gardnerella* biofilm, prophage, endolysin, genus-specificity, antimicrobial resistance, alternative to antibiotic treatment

## Abstract

Bacterial vaginosis is characterized by an imbalance of the vaginal microbiome and a characteristic biofilm formed on the vaginal epithelium, which is initiated and dominated by *Gardnerella* bacteria, and is frequently refractory to antibiotic treatment. We investigated endolysins of the type 1,4-beta-N-acetylmuramidase encoded on *Gardnerella* prophages as an alternative treatment. When recombinantly expressed, these proteins demonstrated strong bactericidal activity against four different *Gardnerella* species. By domain shuffling, we generated several engineered endolysins with 10-fold higher bactericidal activity than any wild-type enzyme. When tested against a panel of 20 *Gardnerella* strains, the most active endolysin, called PM-477, showed minimum inhibitory concentrations of 0.13–8 µg/mL. PM-477 had no effect on beneficial lactobacilli or other species of vaginal bacteria. Furthermore, the efficacy of PM-477 was tested by fluorescence in situ hybridization on vaginal samples of fifteen patients with either first time or recurring bacterial vaginosis. In thirteen cases, PM-477 killed the *Gardnerella* bacteria and physically dissolved the biofilms without affecting the remaining vaginal microbiome. The high selectivity and effectiveness in eliminating *Gardnerella*, both in cultures of isolated strains as well as in clinically derived samples of natural polymicrobial biofilms, makes PM-477 a promising alternative to antibiotics for the treatment of bacterial vaginosis, especially in patients with frequent recurrence.

## 1. Introduction

Bacterial vaginosis (BV) is a very common disorder in women of reproductive age, with a prevalence estimated at 10–30% worldwide [1,2]. It is caused by an imbalance in the normal, healthy microbiome of the vagina that results mostly in discharge, odor, and irritation. BV is associated with an increased risk of preterm delivery and low birthweight [3,4], infertility and early spontaneous abortion [5,6,7], and it is also a high risk factor for contracting sexually transmitted diseases, including HIV [8,9]. Besides these potentially severe physiological consequences, frequently recurring and strongly symptomatic BV can have a huge negative impact on some women’s quality of life and psychological wellbeing.

The healthy vaginal microbiome is characterized by a low diversity and uniform colonization by a few species of *Lactobacillus* [10]. In BV, the imbalance in the vaginal microbiome is represented predominantly by the loss of beneficial lactobacilli and overgrowth of bacteria such as *Gardnerella*, *Prevotella, Atopobium, Sneathia* [11], and *Lactobacillus iners* [12]. The current model of the etiology of BV focuses on the importance of *Gardnerella* species. It postulates that virulent strains of *Gardnerella* form an adherent biofilm on the vaginal epithelium in which other species can proliferate, resulting in a polymicrobial biofilm [13,14,15,16,17]. This adherent biofilm is only weakly affected by the innate immune response [18], and the interplay with the host immune system is only partially understood [19]. BV development may be triggered by sexual transmission of a mix of bacteria including *Gardnerella*, which can adhere to epithelial cells in the presence of lactobacilli. It has been suggested that, unlike strictly anaerobic bacteria associated with BV, *Gardnerella*, a facultative anaerobe, can tolerate the high redox potential created by the Lactobacillus-dominated healthy vaginal microbiome [15,20,21]. The metabolism of *Gardnerella* spp., in turn, results in a local increase in pH and decrease in redox potential, favoring the growth of iron-dependent anaerobes [10] and suppression of lactobacilli [22]. Recently, the species previously known as *Gardnerella vaginalis* was shown to comprise at least 13 different species of which 4 were named [23]. Clinical studies indicated that various *Gardnerella* species may contribute differently to the pathogenesis of BV. A higher abundance of *G. vaginalis* and *G. swidsinskii* was found to be related with vaginal symptoms of abnormal odor and discharge and the relative abundances of *G. vaginalis*, *G. swidsinskii*, and *G. piotii* but not *G. leopoldii* were strongly associated with the BV microbiome [24,25].

The therapies used currently to treat BV are broad-spectrum antibiotics, mainly metronidazole (MDZ), tinidazole (TDZ), or clindamycin (CLI); antiseptics (such as Octenisept^®^, based on the detergent-like octenidin); probiotics (preparations of lactobacilli); and prebiotics (lactate gels). Although all of these treatments may have some beneficial effects, none have proven to have satisfactory efficacy in preventing recurrence. Antibiotics are the most effective in quickly reducing the symptoms, but they are associated with a recurrence rate of up to 60% within six months of treatment [26]. One reason may be the presence of an antibiotic-tolerant biofilm in BV that shelters a reservoir of persister cells [26,27]. Antibiotic treatment not only often fails to eradicate the biofilm, but can also lead to further dysbiosis of the vaginal microbiome and can promote candidiasis [28]. Moreover, CLI can trigger pseudomembranous colitis, cause antibiotics-associated diarrhea, and other gastrointestinal side effects [29,30]. Additionally, *Gardnerella* strains frequently develop resistance to MDZ and TDZ [31]. Antiseptic treatments for BV have been less well studied than antibiotics but they are similar with regard to their effectiveness and are also associated with high recurrence rates and deterioration of the healthy vaginal microbiome [16,32]. Studies of probiotics and prebiotics are generally inconclusive—they do not appear to result in lasting benefits [26]. Vaginal microbiome transplantation after a course of antibiotic treatment had a promising effect in one study of five patients, but this approach is very resource-intense and hardly scalable [33]. In conclusion, novel treatments that offer an alternative to antibiotics or that can be used in combination with antibiotics are urgently needed.

Bacteriophage-encoded peptidoglycan hydrolases—also called endolysins or “enzybiotics”—are a promising alternative to antibiotics [34,35]. Produced towards the end of the lytic cycle in phage-infected bacteria, these enzymes cleave peptidoglycan in the bacterial cell wall, thus lysing the cells and releasing the progeny phages. They have been shown to be particularly effective against Gram-positive bacteria, which lack the Gram-negative outer membrane that might limit accessibility to the cell wall peptidoglycan. Endolysins have several advantages over antibiotics; not least, their very narrow host spectrum, which is usually limited to a single genus or even a single species [36], and their low propensity to generate resistance in their hosts [37]. Bacteriophages that invade Gram-positive bacteria encode a variety of highly diverse endolysins. In general, they have a modular structure consisting of one or more enzymatically active domains (EADs) connected by a flexible interdomain linker to at least one cell wall-binding domain (CBD), typically located at the C-terminus of the protein. Both domains can contribute to the specificity for a given genus or species of bacteria [38].

Although no bacteriophages that infect *Gardnerella* species have yet been isolated, prophages encoding for endolysins are present in the *Gardnerella* genomes. While *Gardnerella* has no outer membrane, it is described as Gram-variable [39], and endolysins for such bacteria have not been described previously. To develop endolysins specific for *Gardnerella*, we used an in silico approach to identify the genes encoding endolysins. We cloned and sequenced 14 such genes, demonstrated the lytic activity and specificity of purified recombinant proteins for *Gardnerella* spp., and engineered an enzyme with enhanced activity by domain shuffling. Moreover, we demonstrate the efficacy and specificity of this engineered endolysin on biofilms of *Gardnerella* in samples from patients with BV.

## 2. Results

### 2.1. Gardnerella Genomes Contain Prophage Genes That Encode Active Endolysins

Although no bacteriophages of *Gardnerella* species are known, multiple *Gardnerella* genomes that have been sequenced contain DNA regions that are predicted to be of prophage origin [40,41]. To identify endolysins that might specifically target *Gardnerella*, we used Basic Local Alignment Search Tool (BLAST) to search for sequences with similarities to known endolysin genes in the prophage regions of published *Gardnerella* genomes. A total of 14 genes were identified that encoded 1,4-beta-N-acetylmuramidases, with sequence identities between 87 and 98% on the protein level, when compared to each other. Twelve genes encode a 306 residues protein, whereas two encode similar proteins that lack 55 residues at the C-terminus (Appendix A). The sequences all encode domain structures common to all known endolysins: an N-terminal EAD of the glycoside hydrolase family 25 (GH25) and a CBD, comprising two CW_7 motifs, homologous to the CBD of Cpl-7 lysozyme, encoded by the *Streptococcus pneumoniae* bacteriophage Cp-7 (Figure 1). The sequences of these putative endolysins are 67–88% identical to some proteins encoded by the genomes of *Atopobium* but less than 70 % identical to any other endolysins, which may already indicate a high specificity for the genus *Gardnerella* (data not shown).

We expressed the 14 predicted endolysin genes (EL1–EL14) as recombinant proteins in *Escherichia coli*, of which ten (EL 1–7 and EL 10–12) yielded sufficient protein to perform further experiments. Their lytic activities were tested on representatives of the four *Gardnerella* species: *G. vaginalis* (ATCC 14018^T^), *G. leopoldii* (UGent 09.48), *G. piotii* (UGent 18.01^T^), and *G. swidsinskii* (GS10234). Bacterial suspensions of 10^7^–10^8^ CFU/mL were incubated with each of the 10 endolysins (at 20 µg/mL) or with buffer as a negative control. After a 5 h incubation, we plated the suspensions and quantified the surviving bacteria, measured as colony-forming units (CFU)/mL (Figure 2). Most of the endolysins reduced the CFU/mL of all four *Gardnerella* species by multiple log_10_ units. Only EL5 and EL6 (one of the two truncated proteins mentioned above) were completely inactive in this assay. EL3 and EL10 were the most active against all four *Gardnerella* species tested, *G. swidsinskii* GS10234 being the most susceptible with a > 5 log_10_-fold reduction in CFU/mL (Figure 2).

### 2.2. Engineering of a Gardnerella Endolysin with Enhanced Bactericidal Activity

To increase the lytic activity of the endolysins identified above, we used domain shuffling [42,43] to systematically recombine ten of the EADs encoded by the newly identified endolysin genes EL 1–7 and EL 10–12 (annotated as H1–7 and H10–12) with nine CBDs encoded by EL1–5, EL7 and EL10–12 (annotated as B1–5, B7, and B10–12; since EL6 was inactive, we did not use its truncated CBD). All of the 81 resulting chimeric proteins were expressed in *E. coli* and purified by means of their N-terminal tags. We tested these chimeric endolysins for their bactericidal activity on the abovementioned *Gardnerella* strains and compared their activity to that of the 10 wild-type endolysins (Table 1). Overall, *G. swidsinskii* GS10234 was most sensitive to most endolysins—wild-type and chimeric. Two of the chimeric endolysins, H2B10 and H2B11, were more active than any wild-type enzyme against all four *Gardnerella* strains, whereas others were more active than any wild-type endolysin against some strains (e.g., H2B12, H5B3, H5B4, H7B3, and H7B11) (Table 1). Overall, chimeric endolysins combining domains H2, H7, or H10 with domains B10, B11, or B12 showed the highest bactericidal activity. The combination of H2 and B10 was most effective against any of the four strains and reduced their viability by an average of 4.3 log_10_ CFU/mL. We designated this construct PM-477 and chose it for further development as a candidate for the treatment of BV.

### 2.3. Specificity of PM-477 for the Genus Gardnerella

We characterized the activity and specificity of the engineered endolysin PM-477 (the combination of the H2 and B10 domain) against panels of 13 *Gardnerella* strains of the 4 species described above, 12 strains of vaginal lactobacilli (including *L. crispatus*, *L. jensenii*, and *L. gasseri,* selected as representative of the healthy vaginal microbiome), and 9 other species typically found in BV, by using the quantitative assay for bacterial viability described above. The engineered endolysin caused a 4–5 log_10_-fold loss of viability in three *Gardnerella* strains (Gs_GS10234, Gs_GS9838-1^T^, Gp_UGent 18.01^T^), a 2–3 log_10_-fold loss of viability in five strains (Gv_ATCC 14018^T^, Gv_BV50.1, Gl_UGent 09.48, Gp_P80275, Gs_BV7.1), and a 1–2 log_10_-fold loss of viability in a further five strains (Gv_UGent 09.07, Gv_UGent 09.01, Gl_BV13.2, Gl_BV86.5, Gp_UGent 21.28) (Figure 3a). The same treatment of lactobacilli, by contrast, had no statistically significant effect on the viability of any of the 12 strains tested (Figure 3b). It also had no effect on strains of *Atopobium vaginae*, *Mobiluncus mulieris*, *Prevotella bivia*, and *Streptococcus agalactiae*, and had only a minor effect on one of the three strains tested of *Mobiluncus curtisii* (Figure 3c). (In the cases of *Atopobium vaginae* and *Prevotella bivia*, the cells either grew poorly or died rapidly upon transfer from agar plates to suspension, even without treatment.) Thus, the engineered endolysin PM-477 is highly selective for *Gardnerella*: it kills strains of each of the four main species [23], without harming beneficial lactobacilli or other species typical of the vaginal microbiome.

The genus-specific bacteriolytic effect of PM-477 was confirmed by microscopy in mixed cultures of *Gardnerella* and lactobacilli. These two genera can be distinguished by their distinct morphologies: *Gardnerella* cells are small coccoidal rods, whereas lactobacilli form mostly long and thick rods. PM-477 (at 460 µg/mL for 5 h) lysed *G. vaginalis* and *G. swidsinskii* cells in monoculture and selectively lysed them in mixed cultures of *G. vaginalis* and *L. crispatus* and of *G. swidsinskii* and *L. gasseri*, respectively, while the lactobacilli in these mixed cultures were unaffected (Figure 4a).

We compared the bactericidal and bacteriolytic effects of PM-477 on *G. vaginalis* and *G. swidsinskii* by treating suspensions of the bacteria with 25–400 µg/mL PM-477 for 5 h and then counting the cell numbers by microscopy and determining their viability by quantitative plating (Figure 4b–e). The highest concentration of PM-477 (400 µg/mL) decreased the number of cell-like structures by only 80–90% (Figure 4b,c), whereas the same treatment resulted in a ≥ 5.6 log_10_ reduction in viable cells. This indicates that PM-477 may kill before fully dissolving bacterial cells, resulting in visible but dead cell-like structures seen under the microscope (Figure 4d,e).

### 2.4. Efficacy of PM 477 Compared to Standard Antimicrobials

To compare the efficacy of PM-477 in suspension to the standard antimicrobials used to treat BV, we determined the minimum inhibitory concentrations (MICs) of PM-477, CLI, MDZ, and TDZ for 20 strains of *Gardnerella* by following the Clinical and Laboratory Standards Institute protocol [44]. All the *Gardnerella* species tested were highly susceptible to PM-477 with a MIC_90_ value (the MIC value which inhibits 90% of strains) of 8 µg/mL (Table 2). The *G. swidsinskii* strains had MICs in the range of 0.25–1 µg/mL, whereas the *G. vaginalis* strains had MICs of 0.13–8 µg/mL, and *G. leopoldii and G. piotii* strains had MICs of 1–8 µg/mL. Among this small number of strains tested for each species, there was no evidence that the four *Gardnerella* species differed with regard to their susceptibility to PM-477. Whereas 12 of the 20 *Gardnerella* strains were resistant to MDZ and 15 were resistant TDZ, CLI was effective against all strains at MICs ≤ 1 µg/mL. All tested strains of three *Lactobacillus* species (*L. crispatus*, *L. gasseri*, and *L. jensenii*) were resistant to the highest concentrations of PM-477, MDZ, and TDZ (MICs > 128 µg/mL), whereas only one strain was resistant to CLI. The concentration which killed >99.5% of cells (minimal bactericidal concentration MBC_99.5_) was also determined and is presented in the Appendix B
Table A1.

### 2.5. Efficacy of PM-477 against Biofilms of Gardnerella in Human BV Samples

To analyze the efficacy of PM-477 in a physiological environment that closely resembles the in vivo situation, we treated vaginal swabs from 15 BV patients, and analyzed them by fluorescence in situ hybridization (FISH).

Areas of strong yellow fluorescence in the untreated (baseline) and buffer control-treated samples indicated the presence of a *Gardnerella*-dominated biofilm on exfoliated vaginal epithelial cells (visible due to their pale autofluorescence and clearly corresponding to DAPI counterstain), called clue cells. At baseline and after buffer treatment, the yellow fluorescence of the *Gardnerella* probe outshines the red fluorescence of the probe for all bacteria, indicating that the biofilm is composed, in large part, of *Gardnerella* cells (Figure 5a). Upon treatment with PM-477 a total resolution of the dense yellow staining on the epithelial cells is seen. Therefore, PM 477 proved strong bacteriolytic activity on *Gardnerella* also in this ex vivo setting. The disappearance of intense yellow fluorescent *Gardnerella* bacteria unmasks the red fluorescence of still intact bacterial mass, indicating mostly non-*Gardnerella* cells. Only very few individual *Gardnerella* cells seemed to remain associated with or in between the epithelial cells. However, the comparison of bacteriolytic and bactericidal effect in Figure 4b–e would indicate that these might be dead cell-like structures with sufficient DNA associated to them to hybridize the FISH probes. In contrast to PM-477, Octenisept killed and lysed both the *Gardnerella* and other bacterial cells non-specifically, as indicated by the visible reduction in both types of fluorescence.

The effect of treatment was also quantified by counting the number of bacterial cells per 10 epithelial cells in representative fields of vision (Figure 5b). When compared with the baseline samples, no significant changes were observed upon incubation with the buffer control, indicating that the vaginal microbiome was generally stable over the time-course of the experiment. PM-477 significantly reduced *Gardnerella* cells in the patient samples (median of 300 cells/10 epithelial cells) when compared to baseline (median of 1200 cells/10 epithelial cells, *p* = 0.02) and when compared to the buffer control-treated samples (median of 900 cells/ 10 epithelial cells, *p* = 0.007): 3 of the 15 samples showed a >500-fold reduction in *Gardnerella* load, 4 showed a >100-fold reduction, 7 a >10-fold reduction, and 11 a >2-fold reduction (data for individual patient samples are presented in Appendix B
Table A2).

The effects of PM-477 on *Gardnerella* were similar to those of Octenisept. Unlike PM-477, however, the lytic activity of the disinfectant was not specific for *Gardnerella* but affected all bacteria (Figure 5a, Appendix B
Table A1). 

Two other potential BV-associated bacteria, *A. vaginae* and *L. iners*, as well as *L. crispatus*, were visualized by using selective FISH probes, and their cell numbers per 10 epithelial cells were counted (Appendix B
Table A2). *A. vaginae*, *L. crispatus*, and *L. iners* were detected in 10, 3, and 8 out of the 15 samples, respectively. Treatment with PM-477 reduced the median cell counts of *L. iners* when compared to treatment with the buffer control in all samples, whereas the numbers of *L. crispatus* and *A. vaginae* bacteria were unaffected. Octenisept strongly reduced the cell counts of all three species.

In summary, these experiments provide proof of the efficacy and selectivity of PM-477 for the genus *Gardnerella* in vitro and also in the native BV biofilm.

## 3. Discussion

This is the first study to show that prophage-derived endolysins active on *Gardnerella* could be used as an innovative treatment for bacterial vaginosis. Although no *Gardnerella* phages were ever isolated in a lab, we were able to identify putative endolysins in silico. We expressed them as recombinant proteins, improved their potency by systematic domain shuffling, demonstrated their efficacy and selectivity in vitro, and, ultimately, on the native polymicrobial biofilms in vaginal swabs from BV patients.

The currently available therapy of bacterial vaginosis is unsatisfactory. Antibiotic-centered “standard-of-care” treatment for BV is associated with a recurrence rate of ~60% within 6 months, which may be explained by the low efficacy of antibiotics on bacteria growing as biofilms, but also by the selective destruction of beneficial lactobacilli, which are thereby prevented from re-establishing a protective environment [26]. Bacteriophage-derived endolysins have been described as a potential alternative to antibiotics; they are both highly active on mucosal biofilms and selective for individual genera or even species of bacteria [35]. Neither endolysins nor whole phages specific to *Gardnerella* have been isolated previously, although the existence of *Gardnerella*-specific phages has been postulated on the basis of genomic analyses [40]. We identified *Gardnerella*-specific endolysin genes in the prophage-like genome sequences of strains of *Gardnerella*. A similar approach was reported earlier, in which the phage lysin PlyCD was identified by searching prophage-like sequences in the genomes of *Clostridium difficile* strains [45]. The fact that it is possible to predict and generate endolysins from prophage encoded sequences in the genomes of unrelated species (*C. difficile* and *Gardnerella*) may indicate a wide application of this approach, which could easily be adapted to other pathogenic bacteria. 

We could show that the wild-type endolysins we identified and, even more so, our engineered constructs, are highly bactericidal and selective for the genus *Gardnerella*. 

Our engineered endolysin PM-477 is superior to the current standard of care antibiotics in that it is active in the low µg/mL range and effective against all 20 of the *Gardnerella* strains we tested, covering all four named species. In total, 60% of the 20 strains were resistant to MDZ and 75% to TZD, reinforcing the notion that PM-477 may be more effective against BV than currently used standard antibiotics.

The high resistance rate for MDZ, which was also reported elsewhere [31], does not support the finding that MDZ treatment temporarily relieves symptoms in a majority of cases [26]. However, long-term cure is mostly not achieved with MDZ, indicating that its effectiveness may be reduced due to the high resistance rate of *Gardnerella* strains [46].

In our study, CLI was also effective against all the strains of different *Gardnerella* genotypes we tested. Curiously, however, CLI has not shown superior efficacy when compared to MDZ in clinical trials [26], which may indicate that BV biofilms are more tolerant to CLI than they are of MDZ, thus compensating the higher rate of resistance against MDZ.

In conclusion, PM-477 is superior to all tested antibiotics because, unlike nitroimidazoles, it is active on all *Gardnerella* strains and, unlike CLI, it does not kill lactobacilli. Antibiotics may have the advantage over PM-477 of being active against a broader spectrum of pathogens. Yet, unlike *Gardnerella*, these pathogens may not be pivotal in vaginal biofilm formation as is the case in BV. Additionally, antibiotics typically have low activity on bacteria in biofilms [27] whereas our ex vivo data indicate that PM-477 works well on *Gardnerella* in vaginal biofilms.

Progress has been made in developing animal models for BV. For example, it was possible to model some of the clinical characteristics of BV, such as epithelial exfoliation and clue cells, upon infection of mice with *Gardnerella* [47]. However, it has not been possible so far to model the increase in pH and the diverse microbiome composition of human BV in animals. Therefore, the next best and, arguably, more biologically relevant model to test the efficacy of PM-477 is to use ex vivo vaginal samples from BV patients, which contain clue cells, i.e., exfoliated epithelial cells covered with a polymicrobial biofilm. The microbial populations of all the ex vivo samples that we collected from 15 individual BV patients were clearly dominated by *Gardnerella* and a substantial part of the *Gardnerella* cells adhered to the epithelial cells. Treatment with PM-477 fully resolved the dense staining due to *Gardnerella* cells on the epithelial cells while not affecting non-*Gardnerella* cells. Thus, we conclude that PM-477 quantitatively removes *Gardnerella* cells inside the polymicrobial biofilm on vaginal epithelial cells ex vivo. Furthermore, it is effective in the environment of the vaginal fluid despite the putative presence of proteases [48] and other substances that might reduce its activity.

PM-477 shows promising results in vitro and ex vivo but we do not know yet whether selectively removing *Gardnerella* will be sufficient to resolve the clinical symptoms of BV, which are caused by a polymicrobial biofilm, and to allow lactobacilli to re-colonize the vagina. The increasing evidence of the central role that *Gardnerella* plays in the etiology of BV [15] and the observation that BV seems not to be possible without *Gardnerella* [22] indicates that this might well be the case. The biomass of the polymicrobial biofilm consists largely of *Gardnerella* cells [13,49], so selective removal of *Gardnerella* should decimate the biofilm. Moreover, *Gardnerella* mediates adhesion to the epithelial cells, whereas other bacteria, especially in smaller numbers, may not be able to initiate biofilm formation on these cells [50]. Other pathogens, especially *Prevotella bivia*, synergize with *Gardnerella*, so when the latter is removed, the viability of the other pathogens is likely reduced as well [51,52]. Removal of *Gardnerella* (and the concomitant reduction in viability of *P. bivia*) would reduce the levels of sialidase produced by *Gardnerella* and *P. bivia* and allow the mucus layer to reform, in turn reducing the adherence of the remaining BV pathogens and depriving them of carbon sources [53]. Finally, the active suppression of lactobacilli by *Gardnerella* [22] would be released, allowing recolonization by benign bacteria. This hypothesis has been investigated with in vitro biofilm models [17,20], and PM-477 as a selective antimicrobial agent may be a very useful tool to further investigate the role of *Gardnerella* in ex vivo as well as in in vivo studies.

In summary, we report here that an engineered, phage-derived endolysin has the potential to completely disrupt the *Gardnerella*-dominated biofilm in vaginal swabs from patients with BV. PM-477 is thus a promising candidate for an effective treatment of recurrent BV, which remains, despite decades of research, a major public health concern.

## 4. Materials and Methods

### 4.1. Bacterial Strains and Culture Conditions

*Atopobium vaginae*, *Gardnerella* spp., *Lactobacillus iners*, *Mobiluncus* spp., *Streptococcus agalactiae*, and *Prevotella bivia* were obtained from the Laboratory of Bacteriology, University of Ghent, Belgium. Other *Lactobacillus* spp. were obtained from the German Collection of Microorganisms (DSMZ, Braunschweig, Germany). *Gardnerella* strains were grown on chocolate (Choc) agar plates (Becton Dickinson, Franklin Lake, NJ, USA) under anaerobic conditions in an anaerobic chamber equipped with anaerobic atmosphere generation bags (Sigma Aldrich, St. Louis, MS, USA) for 24–48 h. *Lactobacillus* spp. were cultured on Schaedler plates supplemented with vitamin K1 and 5% sheep blood (Becton Dickinson), anaerobically at 37 °C for 24–48 h. *A. vaginae, M. curtisii*, *M. mulieris*, *S. agalactiae*, and *P. bivia* were either grown on Columbia blood agar plates, supplemented with 5% sheep blood (Becton Dickinson) or Schaedler agar plates supplemented with vitamin K1 and 5% sheep blood, anaerobically at 37 °C for 48 h. *Escherichia coli* BL21 (DE3) cells (New England Biolabs, Ipswich, MA, USA) were used for protein expression and were grown on Luria Bertani (LB) agar plates (Becton Dickinson) or in LB (Luria/Miller) broth (Carl Roth) supplemented with an appropriate selection antibiotic, aerobically at 37 °C. Liquid cultures of all species were performed in New York City (NYC) broth III (10 mM HEPES (Sigma Aldrich), 15 g/L Proteose Peptone (Sigma Aldrich), 3.8 g/L yeast extract (Thermo Fisher Scientific, Waltham, MA, USA), 86 mM sodium chloride (Carl Roth), 28 mM α-D-glucose (Sigma Aldrich)), supplemented with 10% horse serum (HS) (Thermo Fisher Scientific).

### 4.2. Identification of Gardnerella Genes Encoding Endolysin within Prophage Regions

To identify genes coding for phage endolysins, a comprehensive database of known sequences [40] was blasted against translated nucleotide sequences of all *Gardnerella* genomes entries on NCBI (https://blast.ncbi.nlm.nih.gov). Prophage regions in Gardnerella genomic sequences were identified by using PHASTER (https://phaster.ca/) to scan translated nucleotide sequences of *Gardnerella* genome entries in the NCBI database (https://blast.ncbi.nlm.nih.gov). Protein domain searches for endolysins were performed using Interpro (https://www.ebi.ac.uk/interpro/) to identify predicted enzymatically active domains in the corresponding amino acid sequence for each identified phage endolysin.

### 4.3. Gene Cloning and Overexpression of Phage Endolysins

Briefly, gene sequences encoding wild-type or genetically engineered endolysins were synthesized using codon optimization for *E. coli* (Twist Bioscience, San Francisco, CA, USA), both for the wild-type endolysins and domain-swapped constructs. The DNA constructs were fused to an N-terminal His_6_-tag, cloned into the expression vector pET29b(+) using the Golden Gate system (Thermo Fisher), and expressed in *E. coli* BL21(DE3). Protein expression was induced by TB (Terrific Broth Medium; 24 g/L Bacto Yeast Extract (Becton Dickinson), 12 g/L Bacto Tryptone (Becton Dickinson), 0.4% glycerol (87%; Applichem), 0.02 mM potassium phosphate monobasic (Sigma Aldrich), 0.07 mM di-potassium hydrogen phosphate (Merck, Branchburg, NJ, USA) and 1.5% α-lactose monohydrate (Carl Roth)), by incubating at 25 °C for 24 h, with shaking at 250 rpm/min. The overexpressed proteins were purified by affinity chromatography on a nickel–nitrilotriacetic acid (Ni–NTA) affinity matrix (HISTrap column) eluted with 50 mM MES (Carl Roth) pH 7, 150 mM NaCl (Carl Roth), 250 mM imidazole (Carl Roth) and followed by size exclusion chromatography. Fractions were checked for purity on an SDS-page gel, pooled, and dialyzed against MES buffer (50 mM MES pH 5.5, 100 mM NaCl, 8 mM MgSO_4_ (Sigma Aldrich)). Protein concentration was determined at OD 260/280 nm or by using the Pierce^TM^ BCA (bicinchoninic acid) protein assay kit (Thermo Fisher Scientific) (see Appendix A).

### 4.4. Culture-Based Assessment of Bactericidal Activity

Bacterial suspensions (OD_600_ of 0.1, corresponding to approximately 10^7^–10^8^ CFU/mL) were prepared by scraping the cells from confluently grown agar plates and diluting them into NYCIII+ 10% HS, pH 5.5. Reactions were performed in triplicate by mixing 10 µL of endolysin (200 µg/mL) with 90 µL bacterial suspension in the wells of a 96-well plate. Ten microliters of MES buffer without the endolysin was used as a control. The 96-well reaction plate was incubated anaerobically at 37 °C for 5 h. Tenfold dilution series (10^−1^ to 10^−6^) of the cell reaction mixtures were prepared in NYCB+ 10% HS and 2 µL of each dilution were spotted onto Choc agar plates. After anaerobic incubation at 37 °C for 48 h, colonies were counted, CFU/mL calculated, and the log_10_ reduction compared to MES buffer treated control was determined.

### 4.5. MIC and MBC Assessment

The minimum inhibitory concentration (MIC), which is a standard measure of the activity of antimicrobials, was determined according to the Clinical and Laboratory Standards Institute protocol (2018) *Methods for Antimicrobial Susceptibility Testing of Anaerobic Bacteria* [44]. Bacterial suspensions of 10^5^ to 10^6^ CFU/mL in NYCIII+ 10% HS were treated with a 1:2 dilution series of either PM-477 (starting concentration 64 µg/mL) or the antibiotics metronidazole (Gatt-Koller), tinidazole (Sigma Aldrich), or clindamycin (Clindamycin hydrochloride, Sigma Aldrich), starting with a concentration of 128 µg/mL and twofold dilution down to 0.0625 µg/mL. Controls for growth in the absence of antimicrobials were also included. OD_620_ was recorded by a microplate reader (Tecan, Grödig, Austria) after incubation at 37 °C for 48 h or, for some fast-growing strains, after incubation for 24 h, and compared to *t* = 0 values. Subsequently, to determine the minimum bactericidal concentration (MBC), 2 µL of the antimicrobial dilution series and bacteria suspension of 10^5^ to 10^6^ were spotted on NYC III + 10% HS agar plates and incubated anaerobically for 2–3 days. Colonies were counted and MBC_99.5_ values were determined.

### 4.6. Phase Contrast Microscopy

Dense bacterial suspensions were prepared in NYCIII+ 10 % HS either as *Gardnerella* monocultures or as a co-cultures of *Gardnerella* and *Lactobacillus* strains at final OD_600_= 4 for each strain. Bacterial mixtures were centrifuged at 3800× *g* for 7 min at room temperature in a benchtop centrifuge (Heraeus Instruments, Biofuge Pico). Supernatants were removed and cell pellets were resuspended in 50 µl of PM-477 (460 µg/mL) or in MES buffer alone as a negative control and incubated for 5 h at 37 °C under anaerobic conditions. Subsequently, 5 µL of each bacterial suspension was placed on a glass slide, covered with a glass coverslip, and observed by phase contrast microscopy (Olympus BX41 microscope, magnification 1000×).

For the comparison of bacteriolytic and bactericidal effects, bacterial suspensions of *Gardnerella swidsinskii* and *Gardnerella vaginalis* were prepared in NYCIII+ 10% HS at a final OD_600_ = 1. Bacterial suspensions of 50 µL were centrifuged at 3800× *g* for 7 min at room temperature in a benchtop centrifuge (Heraeus Instruments, Biofuge Pico). Supernatants were removed and pellets were resuspended in 50 µl of MES buffer containing PM-477 at 400, 200, 100, 50, or 25 µg/mL, or in MES buffer alone as a negative control, and incubated for 5 h at 37 °C in anaerobic conditions. Subsequently, 5 µL of each bacterial suspension was observed by phase contrast microscopy as above. To determine viability, 10 µL of the reaction mixture was serially diluted and spotted onto Choc agar plates. After incubation at 37 °C for 48 h, the colonies were counted, and CFU/ml was calculated.

### 4.7. Lytic Effects of PM-477 and Octenisept^®^ on Ex Vivo BV Patient Samples as Detected by Fluorescence in situ Hybridization (FISH) Microscopy

Vaginal samples were collected from randomly selected 24–49-year-old Caucasian women who were previously diagnosed with BV according to the Amsel criteria [54]. In addition, the presence of clue cells (epithelial cells covered with cohesive *Gardnerella* bacteria) was determined by FISH microscopy [16]. Fresh vaginal smears were collected with a swab and put in ESwab™ 493C02 pre-filled vials (COPAN Diagnostics, Murrieta, CA, USA). The vials were vortexed gently, resulting in a suspension that contained epithelial cells and vaginal bacteria. Four consecutive vaginal smears from each patient were pooled. The effects of 0.2, 2, 20, and 200 µg/mL of PM-477 after 2, 6, and 24 h, respectively, were tested either at room temperature or 36 °C. Since the most pronounced lytic effect on *Gardnerella* was observed after 24 h exposure to 200 µg/mL PM-477 at 36 °C (Table A3), this condition was used for further studies. Subsequently, vaginal swabs from 15 BV patients were collected, aliquoted, and mixed 1:1 (*v/v*) with 400 µg/mL PM-477 (final concentration of 200 µg/mL) or with MES buffer as a negative control or mixed 1:20 (*v/v*) with the disinfectant Octenisept^®^ (Schülke, Vienna, Austria) and incubated for 24 h at 37 °C. Untreated samples were also included in the analyses as baseline controls. The samples were fixed with Carnoy solution (alcohol/chloroform/acetic acid 6/3/1 by volume) [55]. Fields of 10 × 10 mm were marked on SuperFrost slides (Langenbrinck, Emmendingen, Germany) with a PAP pen (Kisker-Biotech, Steinfurt, Germany). Five-microliter aliquots of vortexed, fixed vaginal smear sample were dropped onto the marked field. The slides were dried for 60 min at 50 °C before FISH analysis.

For multicolor FISH analyses, we used the Gard662 *Gardnerella*-specific DNA hybridization probe [56] and the universal bacterial probe Eub338 [57]. Oligonucleotide probes were synthesized with a fluorescent dye (Cy3 or Cy5). Hybridization was performed at 50 °C as previously described [55]. The probe Ato291 was used for *Atopobium*, Liner23-2 for *L. iners*, and Lcrisp16-1 for *L. crispatus* [56]. DAPI was used to visualize the vaginal epithelial cells (microphotographs not shown). Changes in density and distribution of microbial species were monitored by using a Nikon Eclipse 80i fluorescence microscope, a Nikon SHG1 camera, and accompanying software (Nikon, Tokyo, Japan). To avoid dispersion-related biases (the thickness of smears on the glass slide and the microbial distribution being uneven), bacteria were enumerated for representative areas containing 10 epithelial cells as follows. For bacteria in low densities (e.g., lactobacilli), larger areas, including at least 10 microscopic fields were evaluated and mean cell numbers were expressed in relation to 10 epithelial cells. In areas with high local bacterial concentrations (e.g., in patches of biofilms), where cells are adjacent and in part overlapping, each 10 × 10 µm area covered with bacteria was counted as 500 bacterial cells. The numbers were rounded to full 100s. 

### 4.8. Statistical Analysis

Where appropriate, data were log-normalized prior to applying statistical tests (e.g., for CFU/mL values, and as indicated in the figure legends). When only two groups were compared, the unpaired two-tailed *t*-test was used as indicated in the respective figure legends. Multiple groups were compared by two-tailed one-way ANOVA tests. The software used for statistical analyses was GraphPad Prism8. Differences between groups were considered statistically significant when *p* ≤ 0.05.

## 5. Conclusions

We report here that an engineered, phage-derived endolysin has the potential to completely disrupt the *Gardnerella*-dominated biofilm in vaginal swabs from patients with BV. PM-477 is thus a promising candidate for an effective treatment of recurrent BV, which is a major public health concern.

## 6. Patents

A patent application under the PCT (PCT/EP2020/062645) resulted from the work reported in this manuscript.

## Figures and Tables

**Figure 1 pathogens-10-00054-f001:**
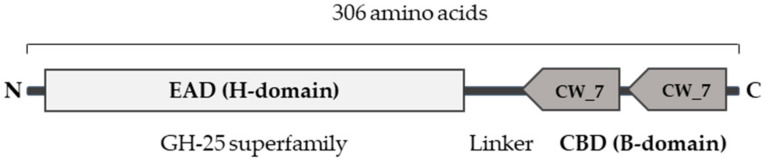
Schematic domain structure of the predicted *Gardnerella* prophage endolysins belonging to the class of 1,4-beta-N-acetylmuramidases. EAD, enzymatic active domain or H-domain; CBD, cell-wall binding domain or B-domain.

**Figure 2 pathogens-10-00054-f002:**
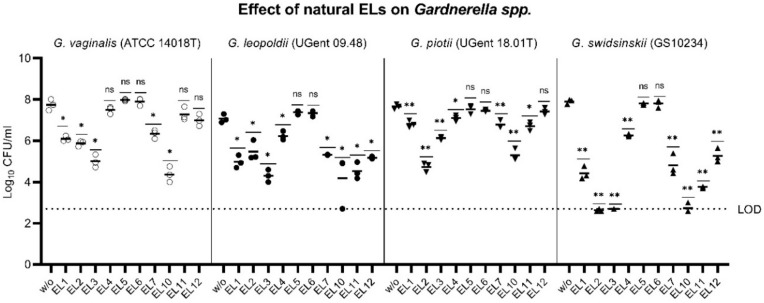
Wild-type endolysins are highly active against four different *Gardnerella* species. Suspensions of *G. vaginalis* (ATCC 14018T), *G. leopoldii* (UGent 09.48), *G. piotii* (UGent 18.01T), and *G. swidsinskii* (GS10234) were treated with recombinant endolysins prepared by expression of the naturally occurring genes (EL1–EL7 and EL10–EL12) (see Methods) and the log_10_ reduction in viable CFU/mL was determined. The log_10_ reduction was calculated by comparing the CFU/mL after endolysin treatment with that of the buffer control (*w/o*). LOD indicates the limit of detection. *p*-values were calculated by one-way ANOVA with multiple comparison to the buffer treated control (*w/o*) on log-normalized data. * *p* ≤ 0.05; ** *p* ≤ 0.01; ns, not significant.

**Figure 3 pathogens-10-00054-f003:**
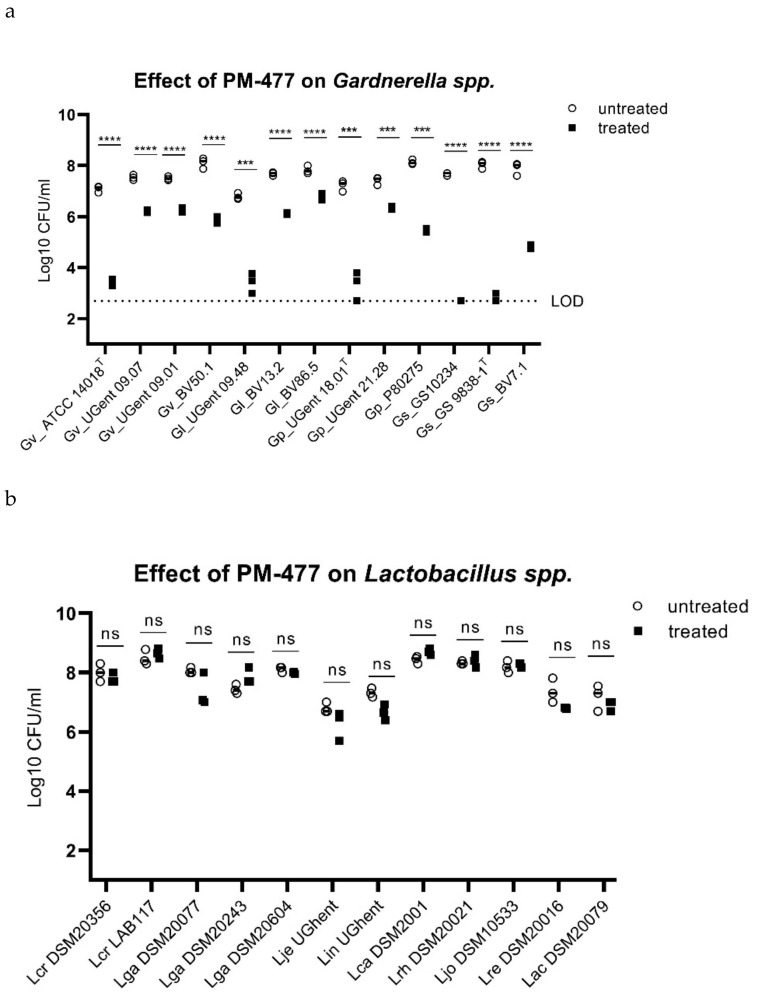
PM-477 is highly specific for *Gardnerella* and spares lactobacilli and other BV-associated microbiome components. (**a**) *Gardnerella* strains, (**b**) *Lactobacillus* spp., (**c**) and *Mobiluncus curtisii*, *Mobiluncus mulieris*, *Streptococcus agalactiae*, *Atopobium vaginae*, and *Prevotella bivia* were tested for their susceptibility to PM-477 (20 µg/mL for 5h; treated, black squares) and compared to buffer control-treated (untreated, white circles) bacteria suspensions. For statistical purposes, CFU/mL values from the untreated and the treated groups were log-normalized prior to applying the unpaired two-tailed multiple Student’s *t*-test. *p*-values ≤ 0.05 were considered statistically significant; ** *p* ≤ 0.01, *** *p* ≤ 0.001, **** *p* ≤ 0.0001.

**Figure 4 pathogens-10-00054-f004:**
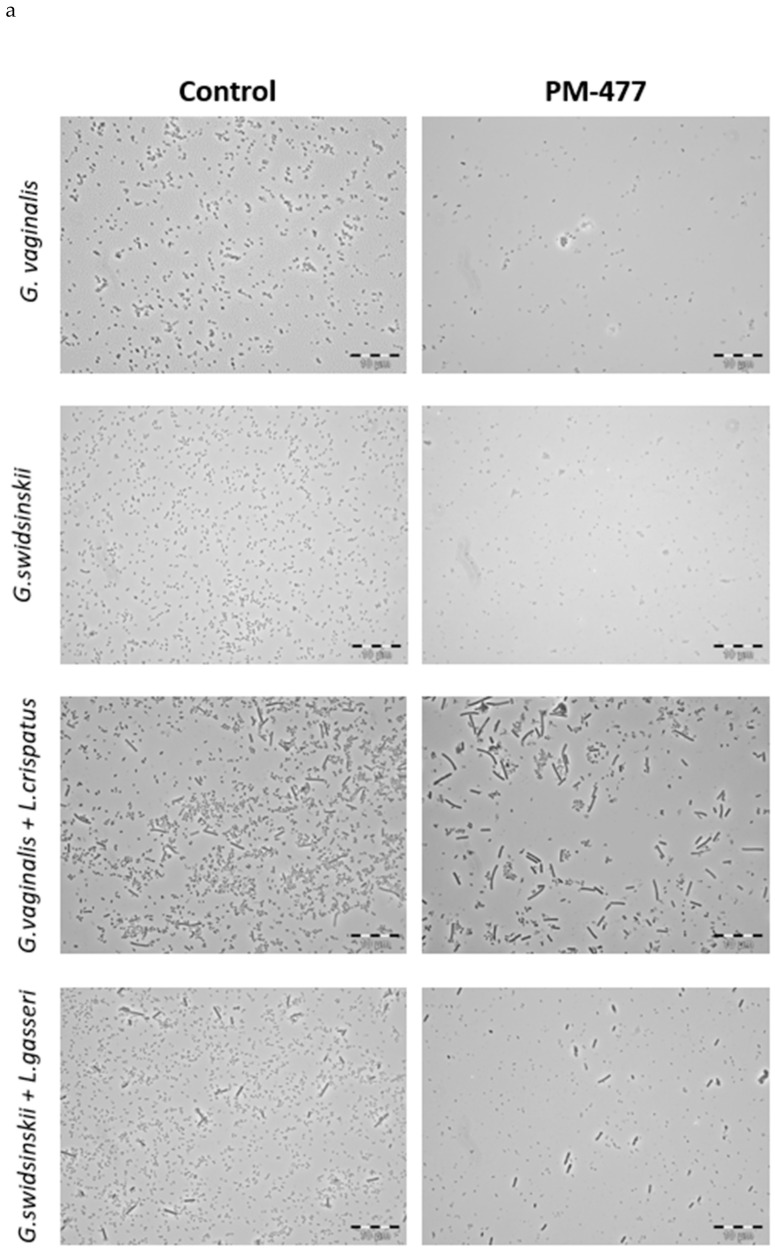
PM-477 selectively kills *Gardnerella* in a co-culture with lactobacilli. (**a**) Phase contrast microscopy of dense cultures of *G. vaginalis* (ATCC14018T) (first row) and *G. swidsinskii* (GS10234) (second row), or co-cultures of *G. vaginalis* (ATCC14018T) and *L. crispatus* (LABCRI FB020-08c) (third row), as well as *G. swidsinskii* (GS10234) and *L. gasseri* (LAB GAS LMG 9203T) (fourth row) treated with PM-477 (460 µg/mL for 5h) or with buffer (control). (**b**,**d**) Cell counts of *Gardnerella* cells determined by phase contrast microscopy after treatment with 25–400 µg/mL PM-477 for 5h or with buffer control (*w/o*). (**c**,**e**) Viable cells remaining (log_10_ reduction CFU/mL compared to the buffer-treated control) in suspensions of *Gardnerella* treated with 25–400 µg/mL PM-477 for 5h. *P*-values were calculated by one-way ANOVA test with multiple comparison to the buffer treated control (*w/o*) on log-normalized data; *** *p* ≤ 0.001, **** *p* ≤ 0.0001.

**Figure 5 pathogens-10-00054-f005:**
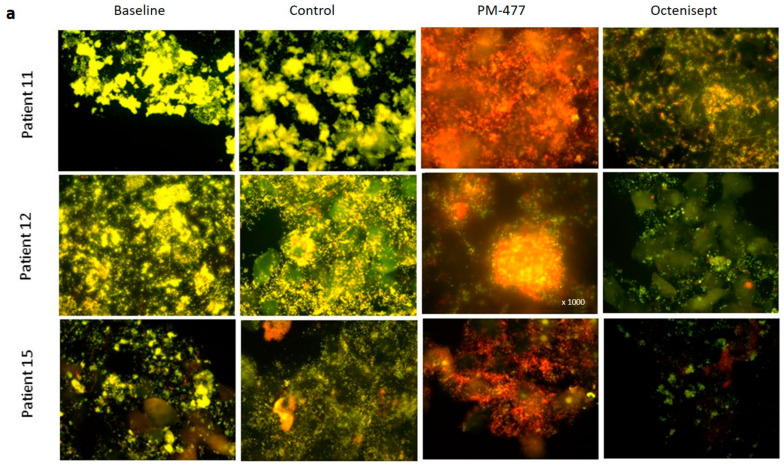
PM-477 dissolves the *Gardnerella*-dominated biofilm on exfoliated vaginal epithelial cells from BV patients. (**a**) Representative examples of multi-color fluorescence in situ hybridization (FISH) with a Gard Cy3 probe (*Gardnerella*, yellow fluorescence) and universal Eub 338 Cy5 probe (all bacteria, red fluorescence) in 400 × magnification (where not stated otherwise) for patients 11, 12, and 15. Baseline (prior to treatment), Control (buffer-treated controls after 24 h incubation), PM-477 (treatment with 200 µg/mL for 24 h), Octenisept (treatment 1/20 for 24 h). (**b**) *Gardnerella* cell density in samples of patients 1–15 expressed as the number of yellow bacteria counted in areas containing 10 epithelial cells (see methods for a detailed description of the counting method). Dotted line represents the lower limit of detection. *p*-values were calculated with one-way ANOVA on log-normalized data in GraphPad Prism; ** *p* ≤ 0.01, *** *p* ≤ 0.001.

**Table 1 pathogens-10-00054-t001:** Domain swapping increases the lytic activity of engineered endolysins. Eighty-one constructs comprising all possible combinations of the EAD (H) and CBD (B) domains of 10 wild-type endolysins were tested on four different *Gardnerella* species as in Figure 2. The values represent the log_10_ of the ratio of viable cells (CFU/mL; mean of triplicate assays) in endolysin-treated suspensions of *Gardnerella* compared to buffer control-treated suspensions of the species, as indicated. The cells are shaded in increasing order of lysis (high reduction in viable CFU/mL), gradually from red (no lysis) over white (moderate lysis) to blue (strong lysis). Wild-type endolysins are boxed in black. The endolysin candidate with the highest average reduction over four *Gardnerella* species is boxed in yellow.

	B1				B2				B3				B4				B5				B7				B10				B11				B12			
	Gv_ATCC 14018^T^	Gl_UGent 09.48	Gp_UGent 18.01^T^	Gs_GS10234	Gv_ATCC 14018^T^	Gl_UGent 09.48	Gp_UGent 18.01^T^	Gs_GS10234	Gv_ATCC 14018^T^	Gl_UGent 09.48	Gp_UGent 18.01^T^	Gs_GS10234	Gv_ATCC 14018^T^	Gl_UGent 09.48	Gp_UGent 18.01^T^	Gs_GS10234	Gv_ATCC 14018^T^	Gl_UGent 09.48	Gp_UGent 18.01^T^	Gs_GS10234	Gv_ATCC 14018^T^	Gl_UGent 09.48	Gp_UGent 18.01^T^	Gs_GS10234	Gv_ATCC 14018^T^	Gl_UGent 09.48	Gp_UGent 18.01^T^	Gs_GS10234	Gv_ATCC 14018^T^	Gl_UGent 09.48	Gp_UGent 18.01^T^	Gs_GS10234	Gv_ATCC 14018^T^	Gl_UGent 09.48	Gp_UGent 18.01^T^	Gs_GS10234
**H1**	−1.9	−1.3	−0.9	−3.5	−0.4	−0.3	−0.2	−1.4	−0.7	−0.7	−0.3	−1.9	−0.8	−0.7	−0.7	−2.3	−1.8	−0.8	−0.8	−3.3	−1.6	−1.2	−0.9	−3.3	−2.5	−2.3	−1.5	−4.9	−1.5	−1.6	−0.8	−3.8	−0.7	−0.6	−0.4	−2.3
**H2**	−1.6	−1.2	−1.0	−3.5	−1.2	−1.1	−2.5	−5.4	−2.9	−2.1	−1.6	−5.4	−2.3	−1.8	−1.6	−4.5	−1.1	−1.5	−2.7	−4.7	−1.0	−1.3	−2.6	−6.7	−3.6	−3.3	−3.8	−6.7	−3.3	−3.0	−3.7	−6.7	−2.8	−3.3	−3.6	−6.7
**H3**	−1.1	−1.3	−2.9	−4.3	−1.6	−1.4	−3.1	−4.1	−3.0	−2.0	−1.6	−5.4	−1.9	−1.4	−1.2	−4.0	−1.4	−1.2	−0.6	−3.5	−0.9	−1.1	−0.3	−2.8	−1.5	−2.2	−0.6	−3.8	−1.6	−2.0	−0.8	−4.2	−1.3	−1.9	−0.5	−3.7
**H4**	0.1	0.7	0.4	0.0	0.1	0.5	0.4	0.0	0.0	0.4	0.2	−0.3	−0.4	−0.5	−0.4	−2.0	0.1	0.3	0.5	0.1	0.1	0.1	−0.4	−1.3	−3.0	−2.9	−1.9	−4.8	−2.9	−3.3	−1.8	−4.8	−0.3	−1.1	−0.3	−3.9
**H5**	−1.3	−1.5	−1.1	−3.8	−1.2	−1.3	−0.9	−3.3	−2.8	−3.0	−1.8	−4.8	−1.8	−3.0	−2.2	−4.5	0.2	0.3	−0.1	−0.1	−1.6	−2.2	−1.2	−4.2	−2.5	−3.6	−1.1	−4.7	−2.4	−1.6	−1.6	−4.4	−2.3	−1.9	−1.6	−4.3
**H6**	−0.3	−0.1	−0.4	−1.8	−0.8	−0.4	−0.7	−2.4	−0.3	−0.5	−0.3	−1.6	−0.4	−0.7	−0.3	−1.6	−0.4	−0.5	−0.4	−1.7	−0.4	−0.8	−0.6	−2.0	−0.5	−0.7	−0.6	−1.8	−1.0	−1.5	−0.7	−2.7	−0.4	−0.6	−0.3	−1.3
**H7**	−0.9	−1.4	−0.9	−3.1	−2.0	−2.2	−1.3	−4.2	−3.2	−3.3	−2.0	−4.8	−2.5	−3.1	−1.7	−4.0	−1.6	−2.1	−1.2	−3.2	−1.6	−2.1	−1.2	−3.2	−3.4	−3.0	−1.8	−5.1	−3.5	−3.3	−1.9	−3.7	−2.6	−3.3	−1.7	−4.1
**H10**	−1.7	−1.1	−1.4	−3.2	−1.6	−1.2	−1.2	−3.9	−2.7	−1.9	−2.1	−4.9	−2.3	−1.5	−1.7	−4.2	−1.2	−0.8	−1.2	−3.5	−1.4	−0.8	−1.2	−3.5	−3.5	−1.6	−2.4	−5.5	−3.2	−1.7	−1.8	−4.6	−3.6	−2.8	−1.6	−5.0
**H11**	−0.8	−1.3	−0.8	−2.3	−0.2	−0.4	−0.4	−0.3	−2.5	−2.2	−1.7	−4.2	−1.7	−1.7	−1.1	−2.8	−1.5	−1.7	−0.9	−3.2	−0.1	−0.9	−0.7	−1.9	−2.8	−2.2	−1.4	−4.1	0.1	−1.6	−1.0	−3.5	−3.0	−2.3	−1.3	−4.1
**H12**	0.0	−0.8	−0.2	−1.5	−0.2	−0.9	−0.3	−2.1	−0.2	−1.2	−0.6	−2.5	0.0	−0.7	−0.3	−2.3	−0.2	−1.1	−0.4	−2.1	−0.8	−0.6	−0.4	−2.6	−1.3	−0.3	−0.4	−2.9	−2.4	−1.0	−1.0	−3.9	−0.2	−1.0	−0.3	−1.9

**Table 2 pathogens-10-00054-t002:** Minimum inhibitory concentration (MIC) of PM-477 and the antibiotics CLI, MDZ, and TDZ for various *Gardnerella* spp. and *Lactobacillus* spp. (R) Resistance, defined as ≥32 µg/mL for MDZ and TDZ, and ≥8 µg/mL for CLI, according to international standards [44].

			MIC of Antimicrobials [µg/mL]			
Species	Identification	Clade	PM-477	CLI	MDZ	TDZ
*G. vaginalis*	ATCC 14018^T^	I	1	0.25	8	128
*G. vaginalis*	UGent 09.07	I	2	0.25	>128	>128
*G. vaginalis*	UGent 09.01	I	0.25	0.13	8	4
*G. vaginalis*	UGent 25.49	I	0.13	<0.06	8	4
*G. vaginalis*	BV50.1	n.d.	8	0.25	32	64
*G. vaginalis*	BV111.5.1	n.d.	2	0.125	8	4
*G. vaginalis*	FB049-01	n.d.	4	0.5	16	8
*G. vaginalis*	FB061-03	n.d.	8	0.25	8	4
*G. leopoldii*	UGent 09.48	IV	4	0.5	128	128
*G. leopoldii*	BV13.2	n.d.	4	0.5	>128	>128
*G. leopoldii*	BV86.5	n.d.	1	0.25	>128	>128
*G. piotii*	UGent 18.01^T^	II	8	0.5	32	64
*G. piotii*	UGent 21.28	II	2	0.25	64	>128
*G. piotii*	P80275	n.d.	1	0.5	16	32
*G. piotii*	FB041	n.d.	4	1	32	64
*G. piotii*	VMF1800 SVT21	n.d.	8	1	32	64
*G. swidsinskii*	GS 10234	IV	0.5	0.25	64	>128
*G. swidsinskii*	GS 9838-1^T^	IV	0.25	<0.06	>128	>128
*G. swidsinskii*	BV7.1	n.d.	1	0.5	>128	128
*G. swidsinskii*	BV112.4	n.d.	1	0.06	64	64
*L. crispatus*	DSM 20584		>128	4	>128	>128
*L. gasseri*	DSM 20077		>128	64	>128	>128
*L. gasseri*	DSM 20243		>128	32	>128	>128
*L. jensenii*	PB2003-073-T2-2		>128	0.25	>128	>128

## Data Availability

Data is contained within the article or Appendix A.

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
