# Peer review of "Engineered Phage Endolysin Eliminates Gardnerella Biofilm without Damaging Beneficial Bacteria in Bacterial Vaginosis Ex Vivo"

_pathogens, 2021, doi:10.3390/pathogens10010054_

Round 1
Reviewer 1 Report
It is a well written manuscript. The topic is important because bacterial vaginosis is a major public health issue and there is no satisfactory treatment yet. Gardnerella vaginalis seems a very good target because it is known that the formation of a polymicrobial biofilm in vaginosis is often initiated by this pathogen.
Authors show robust data to support their conclusions that engineered endolysins from prophages represent a new and efficient strategy against G. vaginalis in vitro and ex-vivo.
Major points
1. The endolysin activity has not been tested against all commensal bacteria. To ensure a little better the specificity of the identified endolysin peptides, a blast of the peptides against predicted endolysins of phages from other bacteria could be performed.
2. Coomassie blue or silver gel staining image should be given showing the purity of purified recombinant endolysins.
Minor issues:
1. G. vaginalis is a Gam-negative bacteria and this is missing in the manuscript. I suggest to add this in the last § of the introduction section.
2. Similarity between 1,4-beta-N-acetylmuramidases should also be given at the peptide level, which is here more important than at the nucleotide level. A comment about the codon usage would be interesting as it can explain the low expression of two recombinant endolysins. In the same line, the authors should consider to add that endolysins of phages that infect Gram-negative bacteria often exhibit a globular structure with a single catalytic domain pointing out that the identified endolysins are not predicted to lyse Garm-positive bacteria.
3. Alignment of peptide sequences should be given outlining the high identity between sequences and showing domains to help the reader.
4. Panels 3b and 3e as 3c and 3d should be side by side to help the reader. Statistic analysis should be added (panels 3c and d)
4. Reference to Table 2 should be added in the Material and Methods section.
5. Fig. 4: two squares should be likely deleted (Octenisept) and the authors should consider to use nonparametric tests here.
6. Check refs. 26, 53 and 57.
Author Response
Thanks for giving us the opportunity to revise our manuscript. All three referees provided very valuable feedback and we are eager to improve our article. Most recommendations were implemented in the revised version of the manuscript. A detailed point-by-point response can be found below.
Responses by Landlinger et al.:
Major points:
- You are right, we claimed that our endoylsins are Gardnerella specific although we tested the specificity only on a panel of lactobacilli and some other bacteria which are associated with the disbalance of the vaginal microbiome, like Mobiluncus curtisii, Mobiluncus mulieris, Streptococcus agalactiae, Atopobium vaginae, and Prevotella bivia. However, we have done extensive BLAST searches and found only high similarities with endolysins within the genome of the family Bifidobacteraceae, the family Gardernella belongs to, 88-67 % identity to some proteins encoded by the genomes of Atopobium, but only < 70 % identity with any other distantly related bacteria. This information is given in the introduction in line 126-127.
- A Coomassie stain and a chromatogram of the recombinantly expressed and purified PM-477 endolysin which was used for the experiments shown in Figure 2-4 and Table 2 was added to the supplementary material (Supplementary Figure 3).
Minor points:
Ad 1: Gardnerella does not consistently retain crystal violet during conventional Gram-staining procedures and often appears as “Gram variable” under the microscope. Gardnerella has a very thin peptidoglycan cell wall and is lacking an outer membrane together with lipopolysaccharide (LPS). Thus, Gardnerella cell walls are unequivocally Gram-positive in their ultrastructural characteristics and chemical composition.
A respective sentence and a reference were added in the introduction in line 104-106.
Ad 2 and 3: We agree, an alignment of the wild-type endolysins would be more conclusive to show the homology of the identified 1,4-beta-N-acetylmuramidases than just the protein sequences listed in a table. We substituted the corresponding table by an alignment of the proteins (Supplementary figure 1). We also indicated the predicted domain structure as requested (see Supplementary Figure 2).
The Supplementary Table 1 was substituted by supplementary Figure 1 and 2.
Ad 4: The graphs have been revised accordingly. The statistical analysis was added as well as the statistical method which was used (see Figure 3).
Ad 4b: Reference describing the resistance breakpoints of clindamycin and metronidazole was added to the table legend.
Ad 5: The squares in Figure 4 did not represent additional data points (they were a graphical artifact created while editing the figure) and were removed as suggested.
Ad 6: These references were checked and revised accordingly.
Additional major changes upon request of other reviewers:
The MBC values were added to the Appendix section. The minimal bactericidal concentrations (MBC 99.5) were already determined for most of the strains but omitted in the first version to not overload the manuscript. We added the data to the Appendix (Table A1) and a respective statement in the text (see line 258-260 and in M&M section 463-466). We defined MBC99.5 as the concentration at which killed 99.5% of cells, starting from a suspension of 105-106 CFU/ml, which is intentionally stringent (compared to the MBC90 definition used in other studies). The highest concentration of PM-477 used in the MBC99.5 study was 54 µg/ml. The MBC99.5 could not be determined on some strains which were particularly hard to culture (e.g. colonies did not grow consistently) and thus we marked them as not determined (n.d.). The MIC table (Table 2) was also slightly revised, as an additional L. gasseri strain was added and the MIC value of one L. jensenii strain for clindamycin was changed from 2 to 0.25 µg/ml, after multiple repetitions of the experiment.

Reviewer 2 Report
The manuscript entitled 'Engineered phage endolysin eliminates Gardnerella in bacterial vaginosis without damaging the healthy vaginal microbiome' addresses an important alternative aspect of antimicrobial control to antibiotics. The authors conducted a very interesting study on the use of recombinant phage endolysins to destroy Gardnerella bacteria in the course of bacterial vaginosis. The methodology used was very innovative and the results obtained were very promising.
Nevertheless, in my opinion, the manuscript needs two important revisions:
- The title is 'Engineered phage endolysin eliminates Gardnerella in bacterial vaginosis without damaging the healthy vaginal microbiome'. This strongly suggests that the authors investigated the vaginal microbiome in BV as well as the effects of PM-477 endolysin. Unfortunately, no in vivo study of the effect of endolysin on the microbiome was conducted, but only on selected bacterial strains in vitro. The title is therefore confusing and must be changed or microbial studies using 16S NGS sequencing must be performed prior to manuscript publication. I recommend the following title 'Engineered phage endolysin eliminates Gardnerella in bacterial vaginosis without damaging the selected representatives of the vaginal microbiome in vitro".
- The authors determined the MIC value. Why was the MBC value not determined? Before publishing an article, it is necessary to provide MBC values.
Yours sincerely,
Author Response
Thanks for giving us the opportunity to revise our manuscript. All three referees provided very valuable feedback and we are eager to improve our article. Most recommendations were implemented in the revised version of the manuscript. A detailed point-by-point response can be found below.
Responses by Landlinger et al.:
- Of course, the reader of this article should not be misled by the title. In our ex-vivo study the endolysin had a strong effect on the Gardnerella biofilm, some effect on Lactobacillus iners, but it did not reduce the cell number of Lactobacillus crispatus or Atopobium vaginae (see Appendix Table A2). These analyses were done by FISH staining of ex vivo vaginal BV samples and testing for more bacterial species was not feasible due to the limited volume of the sample. In vitro, additional bacterial species were investigated for their susceptibility to PM-477. The treatment of lactobacilli had no statistically significant effect on the viability of any of the 12 strains tested (Figure 2b) and no effect on strains of Atopobium vaginae, Mobiluncus mulieris, Prevotella bivia and Streptococcus agalactiae. A minor effect on one out of three strains tested of Mobiluncus curtisii was found (Figure 2c). However, no microbiome sequencing was performed and to make it clearer we agree to change the title of the article (see line 2-4).
- The MBC values were already determined for most of the strains but omitted in the first version to not overload the manuscript. We added the data to the Appendix (Table A1) and a respective statement in the text (see line 258-260 and in M&M section 463-466). We defined MBC5 as the concentration at which killed 99.5% of cells, starting from a suspension of 105-106 CFU/ml, which is intentionally stringent (compared to the MBC90 definition used in other studies). The highest concentration of PM-477 used in the MBC99.5 study was 54 µg/ml. The MBC99.5 could not be determined on some strains which were particularly hard to culture (e.g. colonies did not grow consistently) and thus we marked them as not determined (n.d.). The MIC table (Table 2) was also slightly revised, as an additional L. gasseri strain was added and the MIC value of one L. jensenii strain for clindamycin was changed from 2 to 0.25 µg/ml, after multiple repetitions of the experiment.
Additional major changes upon request of other reviewers:
A Coomassie stain and a chromatogram of the recombinantly expressed and purified PM-477 endolysin which was used for the experiments shown in Figure 2-4 and Table 2 was added to the supplementary material (Supplementary Figure 3).
We agreed with the referee that an alignment of the wild-type endolysins would be more conclusive to show the homology of the identified 1,4-beta-N-acetylmuramidases than just the protein sequences listed in a table. We substituted the corresponding table by an alignment of the proteins (Supplementary figure 1). We also indicated the predicted domain structure as requested (see Supplementary Figure 2). Thus, the Supplementary Table 1 was substituted by supplementary Figure 1 and 2.
Reviewer 3 Report
This paper describes the production and host range of phage-based endolysins against Gardenella
There are major gaps in description of the methods and data used in this paper. These need to be addressed by the authors. Also, there is some problems with the logical presentation. For example, what is PM-477 and how is it related to the other endolysins.
- The species/strain/genome data where the prophage sequences were derived is not sufficiently described. Is this data generated by the authors?
- It is not mentioned how the endolysin genes ended up in plasmids. From PCR-amplified bacterial DNA (which strains)? Or were they synthetic constructs based on blast search?
- Why was the lytic effect measured as change in CFU/ml instead of change in OD?
- The process of genetic engineering has not been explained in methods. It would also be good to add a figure explaining which parts were combined. What kind of cloning techniques were used in producing the chimeric constructs? Or were these synthetic constructs?
- What is PM-477 and how is it related to the endolysins described in previous parts of the manuscript? I assume this is the engineered endolysin, but a sentence actually stating this would be helpful.
Author Response
Thanks for giving us the opportunity to revise our manuscript. All three referees provided very valuable feedback and we are eager to improve our article. Most recommendations were implemented in the revised version of the manuscript. A detailed point-by-point respondse can be found below.
Responses by Landlinger et al.:
Ad 1: The Gardnerella genomes were identified by a comprehensive Blast search of an endolysin library against translated nucleotide sequences of all Gardnerella genomes entries on NCBI. This information was now added in the M&M section in line 419-421.
Ad 2: The genes were synthesized with codons optimized for E. coli based on our Blast and domain search and a more detailed description of the cloning process is given under M&M 4.3.
Ad 3: OD610 measurements were also performed where appropriate (e.g. for MIC determinations). However, in killing assays, OD poorly correlates with the number of live cells (e.g. a 4-log reduction in CFU in Gardnerella corresponded to a only 40 % reduction in OD (data not shown)), probably due to residual turbidity of cell debris. Therefore, we used quantitative CFU plating in most assays.
Ad 4: We added a figure to the supplementary material which indicates the domain structure of the endolysin as well as a multiple alignment of all identified endolysins found on Gardnerella prophage stretches (Supplementary Figure 1 and 2). A more detailed description of the cloning process was also added in the M&M section in line 428-431.
Ad 5: You are right, PM-477 is the candidate that was most effective against any of the four strains Gardnerella strains (Table 1) and is the combination of the hydrolytic H2 and the cell wall binding B10 domain. For better understanding the sentence in line 177-178 was revised.
Additional major changes upon request of other reviewers:
- A Coomassie stain and a chromatogram of the recombinantly expressed and purified PM-477 endolysin which was used for the experiments shown in Figure 2-4 and Table 2 was added to the supplementary material (Supplementary Figure 3).
- We agreed with the referee that an alignment of the wild-type endolysins would be more conclusive to show the homology of the identified 1,4-beta-N-acetylmuramidases than just the protein sequences listed in a table. We substituted the corresponding table by an alignment of the proteins (Supplementary figure 1). We also indicated the predicted domain structure as requested (see Supplementary Figure 2). Thus, the Supplementary Table 1 was substituted by supplementary Figure 1 and 2.
- The MBC values were added to the Appendix section. The minimal bactericidal concentrations (MBC 99.5) were already determined for most of the strains but omitted in the first version to not overload the manuscript. We added the data to the Appendix (Table A1) and a respective statement in the text (see line 258-260 and in M&M section 463-466). We defined MBC99.5 as the concentration at which killed 99.5% of cells, starting from a suspension of 105-106 CFU/ml, which is intentionally stringent (compared to the MBC90 definition used in other studies). The highest concentration of PM-477 used in the MBC99.5 study was 54 µg/ml. The MBC99.5 could not be determined on some strains which were particularly hard to culture (e.g. colonies did not grow consistently) and thus we marked them as not determined (n.d.). The MIC table (Table 2) was also slightly revised, as an additional L. gasseri strain was added and the MIC value of one L. jensenii strain for clindamycin was changed from 2 to 0.25 µg/ml, after multiple repetitions of the experiment.
Round 2
Reviewer 2 Report
Now the manuscript can be published.
Author Response
Dear Reviewer!
Thanks for the acceptance of the manuscript!
On request of Reviewer 3, the schematic drawing of the endolysin structure was revised (EAD was added) and is now part of the main text and presented in Figure 1. Figure numbering was changed accordingly.
Yours sincerely,
Christine (on behalf of all authors)
Reviewer 3 Report
The additions and modifications have clarified the presentation of the manuscript. Especially additions in the supplementary file are very relevant.
However, there is still a bit lack of consistent terminology. I suggest still improving the clarity of the the presentation of the endolysin domains, by clearly indicating the EAD in the graphic presentation in supplementary figure 2. Also, if possible, presenting the supplementary figure 2 in the main text would be very helpful for the reader.
Small suggested edits:
- Line 452: “genes sequences”—> gene sequences
- Line 453: E. coli in italics, also change “natural” to “wild type”
- Tables 2 and 3: check that the fonts and formats match journal settings
Author Response
Dear Reviewer!
Thanks for your valuable feedback.
- The schematic drawing of the endolysin structure was revised (EAD was added) and is now part of the main text, presented in Figure 1. Figure numbering was changed accordingly.
- The typo was corrected, natural was exchanged by wild-type, and E.coli is now written in italic.
- The format of Table 2, A2, and A3 was adapted as requested.
Yours sincerely,
Christine (on behalf of all authors)